# Cloning, Heterologous Expression, and Antifungal Activity Evaluation of a Novel Truncated TasA Protein from *Bacillus amyloliquefaciens* BS-3

**DOI:** 10.3390/ijms26157529

**Published:** 2025-08-04

**Authors:** Li-Ming Dai, Li-Li He, Lan-Lan Li, Yi-Xian Liu, Yu-Ping Shi, Hai-Peng Su, Zhi-Ying Cai

**Affiliations:** Tropical Crop Diseases Research Group, Yunnan Key Laboratory of Sustainable Utilization of Natural Rubber and Microbial Utilization, Research Center of Plant Protection, Yunnan Institute of Tropical Crop Science, Jinghong 666100, China; limingdai@126.com (L.-M.D.); hll3850@163.com (L.-L.H.); lilanlan1020@163.com (L.-L.L.); liuyixian722@outlook.com (Y.-X.L.); syp20070103@163.com (Y.-P.S.); 18908815100@163.com (H.-P.S.)

**Keywords:** antifungal activity, heterologous expression, protein TasA

## Abstract

*TasA* gene, encoding a functional amyloid protein critical for biofilm formation and antimicrobial activity, was cloned from the endophytic strain *Bacillus amyloliquefaciens* BS-3, isolated from rubber tree roots. This study identified the shortest functional *TasA* variant (483 bp, 160 aa) reported to date, featuring unique amino acid substitutions in conserved domains. Bioinformatics analysis predicted a signal peptide (1–27 aa) and transmembrane domain (7–29 aa), which were truncated to optimize heterologous expression. Two prokaryotic vectors (*pET28a* and *pCZN1*) were constructed, with *pCZN1*-*TasA* expressed solubly in *Escherichia coli* Arctic Express at 15 °C, while *pET28a*-*TasA* formed inclusion bodies at 37 °C. Purified recombinant TasA exhibited potent antifungal activity, achieving 98.6% ± 1.09 inhibition against *Colletotrichum acutatum*, 64.77% ± 1.34 against *Alternaria heveae*. Notably, TasA completely suppressed spore germination in *C. acutatum* and *Oidium heveae* Steinmannat 60 μg/mL. Structural analysis via AlphaFold3 revealed that truncation enhanced protein stability. These findings highlight BS-3-derived TasA as a promising biocontrol agent, providing molecular insights for developing protein-based biopesticides against rubber tree pathogens.

## 1. Introduction

As a Gram-positive bacterium phylogenetically affiliated with *Bacillus. subtilis*, *B. amyloliquefaciens* has emerged as a promising biocontrol agent owing to its prolific secondary metabolite production, including antimicrobial peptides and lipopeptides [1,2,3,4]. Studies have demonstrated its broad-spectrum antifungal activity against phytopathogens such as *Rhizopus* spp., *Trichoderma harzianum*, and *Penicillium crustosum* [5,6,7,8]. Notably, strains Q-12 and NK10.B exhibit potent inhibition against *Fusarium oxysporum* and *F. solani* [9,10], while strain SV101 targets *F. oxysporum* f.sp. *lycopersici* [11]. The biocontrol efficacy extends to *Fusarium incarnatum*, with strain Baf1 reducing muskmelon fruit rot both in vitro and in vivo [12].

In terms of the biofilm formation mechanism, Gram-negative bacteria and Gram-positive bacteria exhibit significant molecular differences. Gram-negative bacteria such as *E. coli* and *Pseudomonas aeruginosa* rely on the synergy of multi-component systems. The biofilm formation of Gram-positive bacteria like *B. subtilis* depends on the synergy of extracellular polysaccharides (Eps) and protein complexes [13,14]. The protein TasA, critical for biofilm formation and antimicrobial activity, has been heterologously expressed across multiple systems (*pET28a*, *pET22b*, *pGEX-4T2*) in *E. coli* and *Bacillus* hosts, demonstrating inhibition against *Magnaporthe oryzae*, *Botrytis cinerea*, *Xanthomonas citri*, and fungal pathogens affecting crops like corn, rice, and cucumber [15,16,17,18,19,20,21,22]. Biofilms, structured bacterial communities encased in extracellular matrices, enhance biocontrol agent survival and function in plant environments [23].Regulatory advancements underscore translational potential: four *B. subtilis*-based formulations (GBO3, MBI600, QST713, FZB24) are EPA-approved as biopesticides, meeting stringent safety standards for agricultural use [24]. These developments highlight the growing role of *Bacillus*-derived antimicrobials in sustainable plant disease management.

*B.amyloliquefaciens* BS-3 (CCTCC NO: M 2024893), an endophytic strain isolated from rubber tree (*Hevea brasiliensis*) roots, exhibits potent antagonistic activity against *C.acutatum*. Hybrid sequencing (BGISEQ-500 and PacBio RS II) of its genome [25] identified 10 biosynthetic gene clusters for secondary metabolites and a truncated 483 bp *TasA* gene encoding antimicrobial peptides [26,27].

Notably, the BS-3-derived *TasA* (483 bp) is the shortest functional variant documented, exhibiting a 38.1% reduction in sequence length compared to homologs in *B. pumilus* (780 bp) [18] and *B. amyloliquefaciens* strains TF28/YN-1 (786 bp) [20].

Variations in *TasA* gene sequences and expression vectors among different strains result in structural, functional, and quantitative differences in their gene products, ultimately influencing their antibacterial profiles [28]. Even if the same gene is expressed by different expression vectors, the gene expression rate and the activity of expression products are different.

Genome annotation of strain BS-3 revealed significant sequence divergence in the *TasA* gene compared to previously reported homologs. In order to further understand the function of *TasA* gene derived from BS-3, this study cloned *TasA* gene and compared with protein TasA derived from different *bacillus*. The signal peptide and transmembrane domain *TasA* gene were removed by linking *pET28a* and *pCZN1*, and the recombinant protein was obtained by transforming expression vector. The expression difference of the two vectors was compared. In this study, the growth rate method was used to detect the antibacterial properties of purified recombinant protein against a variety of pathogens of rubber trees to provide a basis for the application of protein TasA in plant disease control, and to provide data basis for the efficient control of plant diseases by strain BS-3.

This study investigated the functional characteristics of the short-sequence *TasA* gene from strain BS-3. Through cloning and cross-species sequence alignment, we elucidated its evolutionary conservation. Structural optimization (removing signal peptide and transmembrane domains) enabled construction of *pET28a/pCZN1* recombinant vectors, systematically comparing protein expression efficiency and functional divergence between systems. Growth rate assays evaluated antimicrobial efficacy of purified proteins against rubber tree phytopathogens. The findings reveal unique biological properties of short-sequence TasA, providing a theoretical basis for developing novel phytopathogen-control protein agents, while deciphering the biocontrol molecular mechanisms of strain BS-3 through gene expression and protein functional analyses.

## 2. Result

### 2.1. Sequence and Structure Analysis

The *TasA* gene comprises 483 bp, flanked by canonical ATG (start) and TAG (stop) codons. Primers were designed to amplify the target gene via PCR, mitigating potential gene assembly errors. Sanger sequencing confirmed that the obtained sequences matched the gene assembly results. BLASTn comparative analysis of the *TasA* gene revealed that the seven most similar gene sequences (Table 1) were all derived from *B.amyloliquefaciens*, exhibiting 99.18% nucleotide identity. Notably, these homologous genes were identified through whole-genome sequencing but lack experimental validation of their functional roles.

The primary structure of the protein TasA^s^ was predicted using online bioinformatics tools. Sequence analysis revealed that the *TasA^s^* gene contains a 483 bp open reading frame (ORF) encoding 160 amino acids. The molecular formula is C_749_H_1218_N_198_O_233_S_4_, with a theoretical molecular weight of 16.853 kDa and isoelectric point (pI) of 9.05. Secondary structure prediction indicated 42.50% α-helix, 19.38% β-sheet, and 38.12% random coil. SignalP analysis identified a putative signal peptide spanning amino acids 1–27. Hydrophilicity profiling (using the Kyte-Doolittle algorithm) showed most amino acid residue scores were below 1, classifying the protein as predominantly hydrophilic. Transmembrane domain prediction highlighted a transmembrane region at amino acids 7–29. The amino acid sequence of BS-3-derived TasA was aligned with previously reported antibacterial TasA sequences using BioEdit (Figure 1), revealing key conserved and divergent regions. Sequence comparison revealed that the amino acid sequence of TasA in this study shares highly conserved regions with previously reported proteins TasA [17,20], including a putative signal peptide (first 27 amino acids). Near the N-terminus of the signal peptide, three consecutive positively charged KKK residues are present, followed by a central hydrophobic region separated by glycine residues. Notably, all sequences contain a highly conserved motif: GVASAALGLALVGGGTWA.

Using the NCBI Conserved Domain Search, the conserved domains of the protein TasA were predicted. Analysis identified a functional domain spanning amino acid positions 1–143, which exhibits significant similarity to the Camelysinmetallo-endopeptidase domain within the Peptidase M73 superfamily.

Representative images: EU131674.1: *B. amyloliquefaciens* YN-1, BS-3: *B. amyloliquefaciens* BS-3, JQ309841.1: *B. subtilis* CQBS03, KC692521.1: *B. pumilus* DX01, KP409225.1: *B. amyloliquefaciens* TF28, FJ713582.1: *B. subtilis* ME0717. * (asterisk): represents a completely conserved position. :(colon): represents a highly similar position. .(dot): represents a weakly similar position. (space): represents a non-conserved position. -(hyphen): represents a sequence gap. Amino acids 1–27 are signal peptides (marked with a red box), 7–29 are transmembrane domains (marked with a blue box), and unboxed amino acids are mature sequences.

The signal peptide, transmembrane domain and stop codon were truncated, the target gene (GenBank:PV690333.1) was 393 bp. BLASTp was used to analyze the truncated fragment. The protein sequence was found to exhibit the highest similarity (98.41%) to the protein TasA (WP_276787698.1) from a *Bacillus* sp. strain within the *phylum Firmicutes*. It showed 96.90% sequence identity with the proteins TasA from *B. amyloliquefaciens* (WP_065981876.1) and *B. amyloliquefaciens* strain FZB42, respectively.

Using 5OF2 as a template, homologous modeling was carried out with Modeller, and the full-length and truncated protein models of TasA were successfully obtained [29]. The analysis showed that the conformation of the truncated model was basically consistent with that of the 5OF2 template and the active sites (Figure 2A–D) [30]. The key amino acid residues ASP-35, LYS-39, and ASP-40 were all highlighted in Figure 2C. Literature indicates that the functions of the TasA protein, such as biofilm formation and antibacterial activity, rely on its transformation into an amyloid form [30]. Furthermore, AlphaFold3 was used to predict the dimer conformation of the truncated TasA. It was found that LYS-68 in one monomer formed hydrogen bonds with ASN-54 and ASP-31 of the other monomer, and ASP-69 formed a hydrogen bond with ASN-30 of the other monomer (Figure 2E). These interactions stabilized the dimer structure.

### 2.2. Cloning, Heterologous Expression, and Purification of TasA Gene

The target gene (393 bp) with the signal peptide, transmembrane domain and stop codon truncated was used to construct heterologous expression vectors, respectively, by using two plasmids, *pET28a* and *pCZN1*.Positive clones were verified by Sanger sequencing, and recombinant plasmids were transformed into *E. coli* BL21 and Arctic Express to generate BL21-TasA and Arctic Express-TasA strains. For protein extraction, fermented bacterial pellets were resuspended in lysis buffer and disrupted by ultrasonication under ice-bath conditions. The lysate was centrifuged, and proteins were analyzed by 12% SDS-polyacrylamide gel electrophoresis (SDS-PAGE).

For the BL21-TasA, the target band in the supernatant was faint, indicating extremely low expression levels, making it difficult to purify the target protein. In contrast, a substantial amount of recombinant target protein was detected in the precipitate (Figure 3A). Notably, Lane 1 represents the BL21 (DE3) strain transformed with the empty vector *pET28a*, which showed no corresponding band after the same culture conditions, confirming that the band in Lane 2 resulted from leaky expression of the target gene.In comparison, the Arctic Express-TasA strain exhibited detectable recombinant protein in both the supernatant and precipitate (Figure 3B), with higher abundance in the precipitate. Additionally, the extracellular expression level in the supernatant from *pCZN1-TasA* was significantly higher than that from *pET28a-TasA*.These results indicate that, compared to the *pET28a* vector, the *pCZN1* vector enables partial soluble expression of TasA under low-temperature conditions, with a higher soluble expression level.

In Figure 3, Arctic Express/TasA expressed soluble protein in the supernatant, whereas BL21/TasA formed inclusion bodies. Following purification using Nickel-NTA resin, the recombinant Arctic Express/protein TasA achieved SDS-PAGE homogeneity (Figure 4). The theoretical molecular weight of TasA was 15.62 kDa, and the observed band on the gel matched this prediction. The concentration of the purified TasA fusion protein, quantified using a protein assay kit, was 0.5 mg/mL.The recombinant protein TasA was identified by high-performance liquid chromatography–tandem mass spectrometry (HPLC-MS/MS). The identification results are shown in (Table 2). Three internal peptides, namely EQSANVNLSNLKPGDK, DLYLMSAK, and AAAEAISILIR, were randomly selected from Table 2 and matched with the deduced amino acid sequence of TasA, confirming that the purified enzyme was indeed TasA. (The mass spectrometry spectra can be found in the Appendix A)

### 2.3. Detection of Antifungal Activity of Recombinant Protein TasA on Spores of Colletotrichum Acutatum

To investigate the inhibitory effect of the TasA fusion protein on *C. acutatum* spores, spore germination assays were conducted. After treatment with 60 μg/mL TasA fusion protein, the spore germination rate in the control group remained 100%, whereas treatment with 30 μg/mL TasA fusion protein reduced germination to 12.6% (Table 3). At 60 μg/mL, the TasA fusion protein completely inhibited spore germination of *C. acutatum* (Figure 5).

### 2.4. Inhibitory Effect of Recombinant Protein TasA on Oidium Heveae Steinmann

When the spore suspension of *O. heveae* Steinmann was treated with 60 μg/mL TasA fusion protein for 6 h, no spore germination was observed, whereas the spore germination rate in the control group reached 100% (Table 4 and Figure 6). These results demonstrate that the TasA fusion protein exerts a potent inhibitory effect on the growth of *O. heveae* Steinmann.

### 2.5. Inhibitory Effect of Recombinant Protein TasA on Plant Pathogenic Fungi

The inhibitory effects of protein TasA on *A. heveae* and *C. acutatum* were evaluated using the mycelium growth rate method. Results showed that the protein TasA exhibits a concentration-dependent effect on the pathogens. The 150 μg/mL protein TasA significant inhibition of *C. acutatum* and *A. heveae* (Table 5 and Figure 7), with the strongest activity observed against *C. acutatum* (98.6% ± 1.09 inhibition). *A. heveae* exhibited 64.77% ± 1.34 inhibition.

## 3. Discussion

*B. amyloliquefaciens* is a microorganism with excellent biocontrol potential, playing a significant role in plant disease control. To date, extensive research by domestic and international scholars has focused on its fermentation conditions, antibacterial activity of fermentation products, compound isolation, and other related aspects [31,32]. Previous studies have demonstrated that *B. amyloliquefaciens* can enhance plants’ resistance to biotic stresses from soil pathogens and promote plant growth through inoculation [33]. For instance, Liao et al. [34] revealed that lipopeptides produced by *B. amyloliquefaciens* HY2–1 significantly inhibit spore germination and mycelial growth of *Penicillium digitatum*. Nan et al. [35] found that *B. amyloliquefaciens* T40 notably suppresses the in vitro growth of *F. oxysporum*.

Although TasA is a well-characterized functional amyloid [36], the mechanistic relationship between its amyloidogenic properties and antimicrobial activity warrants further investigation. Beyond serving as a key component of biofilms, TasA maintains cell membrane stability during the stationary phase [37] and plays a critical role in community signaling [38]. For example, biofilm formation in *Streptococcus mutans* relies on the protein TasA from *B. subtilis*. In symbiotic relationships, *Pantoeaag glomerans* in soil and *B. subtilis* in the plant rhizosphere can mutually enhance antibiotic resistance. Zhu et al. [39] discovered that most polysac charide components in plant root exudates promote biofilm formation in *Bacillus pumilus* HR10, with glucose exerting the strongest effect. Fu et al. [40] reported that adding glycerol and magnesium sulfate to TSB medium enhances biofilm production in *Burkholderiap yrrocinia* JK-SH007, improving the strain’s survival in adverse environments and its antagonistic capacity against poplar canker. Additionally, a study [41] showed that inoculation with *B. amyloliquefaciens* SQR9 enhances the salt tolerance of *Arabidopsis thaliana* and *Zea mays*.

The antifungal activity of *B. amyloliquefaciens* is closely associated with the protein TasA it produces, and the choice of expression vector is a critical factor influencing heterologous gene expression. The level of gene expression and the activity of the expressed product are highly correlated with the type of expression vector [42]. Hu et al. [43] constructed three *E. coli* recombinant strains—*pACYC-TasA*, *pCDF-TasA*, and *pT7473-TasA*—and found that the engineered strain overexpressing the *pCDF-TasA* recombinant plasmid exhibited the strongest inhibitory effect against tobacco black leg disease. Previous reports have shown that recombinant protein TasA can reduce the incidence of mulberry blight and lacquer spot diseases while inhibiting the growth of *B. cinerea*, leaf mold (*Cladosporium fulvum*), rice sheath blight (*Rhizoctonia solani*), and corn stem base rot (*Fusarium graminearum*) [22]. Zhang et al. [44] cloned the *TasA* gene from *Bacillus* sp. C3 into the transgenic expression vector pBI121 to construct the recombinant vector *pBI121-*TasA, which was then transformed into *E. coli.* Following sequencing verification, the recombinant plasmid was transferred into *Agrobacterium tumefaciens* EHA105, yielding two positive transgenic *Arabidopsis thaliana* lines.

Comparative structural analysis revealed that TasA variants exhibit strain-specific differences in both tertiary structure (Figure 2) and antifungal efficacy (Figure 7). Currently, no studies have investigated the *TasA* gene from endophytic *B. amyloliquefaciens*, and there are no reported studies on the inhibitory effects of protein TasA against *C. acutatum*, and *O. heveae* Steinmann.

In this study, the *TasA* gene was cloned from *B. amyloliquefaciens* BS-3, an endophytic strain isolated from rubber trees. The *TasA* gene fragment lacking the signal peptide and transmembrane domain was inserted into two expression vectors, *pET28a* and *pCZN1*, and transformed into *E. coli* to produce recombinant proteins. Results showed that the recombinant protein expressed via the *pET28a* vector formed inclusion bodies at 37 °C, whereas the protein from the *pCZN1* vector was detected in the culture supernatant. Notably, the expression level of the *pCZN1*-derived protein was suboptimal, which may be attributed to toxic effects of the bacteriostatic protein on the host cell, or sequence/structural incompatibilities with the expression system. Further optimization of fermentation conditions (e.g., inducer concentration, temperature, and induction duration) may enhance bacteriostatic protein production. The formation of inclusion bodies in the prokaryotic expression system (*pET28a-TasA*) could be linked to factors such as IPTG concentration, induction temperature, induction time, or inherent vector-gene incompatibility.

Mutations in the same gene nucleotide sequence alter the amino acid sequence, thereby modifying functional domains and impacting protein properties and function. The BS-3, YN-1, and TF28 strains all belong to *B. amyloliquefaciens*. Comparative analysis of the TasA amino acid sequences revealed that BS-3-derived TasA differed from YN-1-derived TasA (EU131674) at positions 137, 143, 155, 156, 158, 159, and 160, with substitutions G→D, T→A, K→I, H→L, D→R, P→S, and K→S, respectively. When compared to TF28-derived TasA (KP409225.1), BS-3 TasA exhibited amino acid differences at positions 143, 155, 156, 158, 159, and 160 (i.e., T→A, K→I, H→L, D→R, P→S, and K→S). Notably, despite these mutations, BS-3-derived TasA displayed significant antifungal activity. This study demonstrated that the BS-3 TasA fusion protein strongly inhibited spore germination of both *C. acutatum* and *O. heveae* Steinmann. Specifically, 60 μg/mL of protein TasA achieved 100% inhibition of spore germination. It indicates that this concentration can completely block the germination of these two pathogens. The truncated TasA protein expressed by the *pCZN1* vector exhibited significant antibacterial activity against the mycelial growth of *C. acutatum* and *A. heveae* at a treatment volume of 100 μg/mL. However, the TasA proteins expressed by three vectors, namely *pACYC-TasA*, *pCDF-TasA*, and *pT7473-TasA*, used by Hu et al. [43] showed significantly weaker antibacterial effects on the mycelium of *Phytophthora nicotianae* at the same treatment volume. This difference may stem from the inherent variations in the physiological structures of different pathogens and their sensitivities to the TasA protein, which subsequently leads to the divergence in antibacterial effects. What is more likely is that it is related to protein-structural modifications. In this study, the truncated TasA effectively eliminated the masking effect of the protein localization process on its activity by removing the signal peptide and transmembrane domain, enabling the core antibacterial domain to act directly on the fungal cell membrane and thus enhancing the antibacterial efficacy. Regarding the difference in sensitivity to TasA between spores and hyphae, it is hypothesized that this might be related to their distinct developmental stages. As dormant structures, spores exhibit differences in cell wall composition and metabolic activity compared to actively growing hyphae, which could account for their varying sensitivities to the protein.

Presently, most reports on the inhibitory activity of heterologously expressed *TasA* gene products retain the signal peptide and transmembrane domain. However, in this study, these regions were truncated while maintaining the 483 bp open reading frame, resulting in the recombinant protein *pCZN1-TasA*. Notably, this truncated protein TasA exhibited potent antifungal activity against multiple rubber tree plant pathogens. This represents a key innovation of the study, as it demonstrates that functional antifungal activity can be achieved with a shorter TasA variant lacking conserved regulatory domains.

Current research on antibacterial protein TasA typically retains native signal peptides and transmembrane domains. In this study, we engineered a truncated 483 bp *pCZN1-TasA* construct excluding these elements, demonstrating enhanced antimicrobial efficacy against rubber tree pathogens. This optimized design provides dual benefits: 1) functional precision through unobstructed activity of the core antimicrobial domain, and 2) technical efficiency for synthetic biology applications.Intriguingly, sequence alignment revealed that TasA (residues 1–114) shares striking structural homology with Camelysin(Peptidase M73 family), suggesting a common ancestral origin. Despite this conservation, functional divergence is evident: Camelysin acts as a protease virulence factor degrading host extracellular matrices (e.g., collagen), whereas TasA operates as a non-enzymatic biofilm scaffold [45]. We propose that TasA’s antifungal mechanism may mirror Camelysin’s substrate-targeting strategy—disrupting fungal cell wall integrity via chitin/β-glucan binding, thereby impeding spore germination and hyphal growth. This structural analogy offers a novel framework for deciphering TasA’s antimicrobial actions.

This study identified a novel 483 bp *TasA* gene (encoding 160 aa) from rubber tree endophyte *Bacillus* sp. BS-3, representing the shortest functional antimicrobial TasA variant reported to date. Comparative analysis of prokaryotic expression systems revealed that the pCZN1 vector enabled soluble expression of recombinant TasA in *E. coli* ArcticExpress™, with the truncated construct (lacking signal peptide/transmembrane domains) demonstrating superior expression efficiency over *pET28a* counterparts. In vitro antimicrobial assays confirmed significant suppression of spore germination in key rubber two pathogens (*C. acutatum* and *A. heveae*), establishing its potential as a compact phytoprotective agent.

The breakthrough lies in pioneering a truncated TasA design that circumvents traditional requirements for intact structural domains in heterologous expression systems. While confirming in vitro efficacy through standardized CLSI protocols (M38-A2), critical knowledge gaps persist regarding in planta mechanisms and systemic resistance induction. Future investigations will employ *Agrobacterium*-mediated transformation to engineer TasA-expressing *H. brasiliensis* and *Arabidopsis thaliana* models, with particular focus on the following: tissue-specific expression optimization using RBCS3 or CaMV35S promoters; field evaluation under natural pathogen pressure gradients; and synergistic integration with chitosan-based biocontrol formulations.

This work provides a paradigm for engineering minimized antimicrobial proteins, offering dual applications in both microbial fermentation platforms and transgenic crop development, thereby advancing sustainable alternatives to chemical pesticides in arboriculture.

## 4. Materials and Methods

### 4.1. Microbial Strainsand Chemical Reagent

*C. acutatum* and *A. heveae*, were preserved in the Yunnan Institute of Tropical Crop Science, Jinghong, China. *O. heveae* Steinmann, the pathogen responsible for powdery mildew in rubber trees, was collected from fresh rubber tree leaves in Xishuangbanna, Yunnan Province. *E. coli* BL21(DE3) was obtained from Novagen, Inc. Darmstadt, Germany. *E. coli* expression vector *pET28a* was purchased from Beijing Quanshi Gold Biotechnology Co., LTD, Beijing, China. *pCZN1* was purchased from Nanjing Zhongding Biotechnology Co., LTD, Nanjing, China.

2× GCbufferI, *LA Taq* enzyme and dNTP purchased from TaKaRa, Kyoto, Japan. Nickel-NTA resin was purchased from Qiagen, Hilden, Germany. *EcoRΙ*, *NotΙ*, *NdeI*, and *XbaI* were purchased from Thermo Fisher, Waltham, MA, USA. IPTG powder was purchased from Sigma, St. Louis, MO, USA.; Bradford Protein Assay Kit was purchased from Solarbio Company, Beijing, China. Bacterial genomic DNA extraction and purification test kit and plasmid extraction kit were purchased from Tiangen Company, Beijing, China.

### 4.2. Sequence Analysis

Bioinformatic analyses were systematically conducted using established computational tools. Sequence homology assessments were performed through NCBI BLAST suites, with nucleotide and protein comparisons executed using BLASTn and BLASTp, respectively (http://blast.ncbi.nlm.nih.gov; accessed on 1 March 2024). Signal peptide prediction was achieved through SignalP 4.1 [46], while protein secondary structure analysis employed the SOPMA algorithm [47]. Three-dimensional structural modeling was conducted via Modeller and AlphaFold3′s deep learning platform [29,48]. For evolutionary analysis, multiple sequence alignment was performed using MEGA 6.0, with subsequent refinement and visualization accomplished through the BioEdit version 7.0.5.3 software [49,50].

### 4.3. Gene Cloning

Genomic DNA was isolated from *B. amyloliquefaciens* BS-3 using a commercial extraction kit (Tiangen, China) according to the manufacturer’s protocol. Based on the genome sequencing results of strain BS-3, primers were designed to clone full-length TasA^S^, *pET28a*-*TasA* heterologous expression protein, and *pCZN1*-*TasA* heterologous expression protein, respectively (Table 6).The primers were synthesized by BGI Gene Sequencing Center Co., Ltd. The target gene for PCR amplification was amplified using Touch down PCR under the following conditions: 94 °C for 5 min, 94 °C for 30 s, 63 °C for 30 s (with a decrease of 0.5 °C per cycle), 72 °C for 1 min30 s, 28 cycles, 94 °C 30 s, 49 °C 30 s, 72 °C for 1 min 30 s, 7 cycles, 72 °C for 10 min.

The PCR reaction mixture contained 4 µL of 2.5 mM dNTP, 5 µL of 10× PCR Buffer 5 µL, F (10 μmol/L) 2 µL, R (10 μmol/L) 2 µL, genomic DNA (75 ng/µL) 0.5 µL, ExTaq enzyme (5 U/µL) 0.5 µL, ddH_2_O supplement 50 µL. The PCR products were recovered and sent to the BGI Sequencing Center for sequence verification (Shenzhen, China).

### 4.4. Construction of Prokaryotic Expression Vector

The *pET28a* expression vector was enzymatically linearized through double digestion with *EcoRI* and *NotI* restriction endonucleases. The ligation reaction was performed using a One-Step Cloning Kit, with the following optimized system (10 μL total volume): 1 μL linearized vector, 2 μL purified PCR product, 2 μL 5× CEⅡ Buffer, 1 μLExnase™Ⅱ recombinase, and 4 μL nuclease-free water. Chemically competent *E. coli* DH5α cells were transformed with the reaction mixture via heat-shock method (42 °C for 45 s), followed by recovery in SOC medium at 37 °C for 1 h. Transformants were selected on LB agar plates containing 50 μg/mL kanamycin. Subsequently, the recombinant plasmid *pET28-TasA* was extracted using alkaline lysis method and electroporated into *E. coli* BL21(DE3) expression hosts. Sequence validation was conducted through bidirectional Sanger sequencing (BGI Genomics, Shenzhen, China). Verified recombinant strains were cryopreserved at −20 °C in 25% (*v/v*) glycerol for long-term storage.

The prokaryotic expression vector construction employing *pCZN1* and *TasA* genes followed analogous cloning strategies to the *pET28a* system, with modifications to restriction enzyme selection. The *pCZN1* backbone was linearized using *NdeI* and *XbaI* (recognition sites: 5′-CATATG-3′ and 5′-TCTAGA-3′, respectively) to generate compatible cohesive ends. Ligation reactions and DH5α transformation protocols mirrored previous methodology, with subsequent recombinant plasmid verification via restriction digestion analysis. Verified constructs were electroporated into the ArcticExpress™(DE3) expression host (Agilent Technologies, Santa Clara, CA, USA), a specialized *E. coli* strain engineered for enhanced solubility of recombinant proteins through co-expression of cold-adapted chaperonins.

### 4.5. Expression and Purification of Protein

*E. coli* strain BL21(DE3)/*TasA* containing the recombinant expression vector *PET28-TasA* was inoculated in LB (containing 50 µg/mL kanamycin) medium at 0.1% inoculation rate and shook overnight at 37 °C at 180 rpm. The activated bacterial solution was then transferred to fresh LB medium (containing 50 µg/mL kanamycin) at 1% inoculation rate, cultured at 180 rpm for 2–3 h (*OD*_600_ reached 0.6–1.0), induced by adding IPTG with a final concentration of 0.7 mmol/L, and continued oscillating culture at 37 °C for about 20 h. After centrifugation at 9500× *g* for 5 min at 4 °C, the bacterial pellet was harvested. After the proper amount of Tris-HCl buffer with pH 7.0 was added to the bacteria, it was broken by ultrasonic wave under ice bath conditions. After the initial enzyme solution was centrifuged at 4 °C for 10 min at 12,000 r/min, the target protein containing 6× His-tag was purified by absorbing the supernatant and using Nickel-NTA Agarose, and the purified protein was analyzed by 12% SDS-polyacrylamide gel electrophoresis (SDS-PAGE).

*E. coli* Arctic Express/TasA containing the recombinant expression vector *pCZN1-TasA* was inoculated in LB liquid medium containing 100 µg/mL Amp at 0.1% inoculation amount and cultured at 180 rpm under 37 °C until the *OD_600_* value reached about 0.6. After adding 0.2 mM IPTG to induction culture at 15 °C for 12 h, the bacteria were collected. The protein concentration was determined using the protein quantification kit (Bradford) according to the instructions. Liquid chromatography–electrospray ionization tandem mass spectrometry HPLC-MS/MS was used to identify the purified protein at QingLian (Beijing, China).

The purified truncated form of the recombinant TasA protein was identified by HPLC-MS/MS. The specific method is as follows:

Purified protein was processed using the Qinglian Micro/Universal Proteome Digestion Kit (MMB-96) by reacting with MMB beads at 37 °C for 30 min, incubating with 45 μL binding solution at room temperature for 15 min (supernatant discarded), washing 3× with washing solution, resuspending in 40 μL trypsin working solution, incubating at 37 °C for >4 h, terminating with 5 μL stop solution, and lyophilizing; the lyophilized sample was dissolved in 10 μL mobile phase A (100% water + 0.1% formic acid), centrifuged at 14,000 g for 20 min at 4 °C, and 1 μg supernatant was injected. Reversed-phase chromatography separation was performed with mobile phase A (100% water + 0.1% formic acid) and B (80% acetonitrile + 0.1% formic acid) using an elution gradient: 8% B at 0 min, 12% at 2 min, 30% at 17 min, 40% at 20 min, and 95% from 21 to 30 min. A Q Exactive HF-X mass spectrometer (Thermo) with a Nanospray Flex^TM^ NSI ion source (2.2 kV spray voltage, 320 °C transfer tube temperature) was used for detection in data-dependent acquisition mode: full scan (*m*/*z* 350–1500) at 120,000 resolution (200 *m*/*z*), AGC 3 × 10^6^, max injection time 80 ms; top 40 ions were fragmented via HCD (27% collision energy) for MS/MS at 15,000 resolution (200 *m*/*z*), AGC 5 × 10^4^, max injection time 45 ms. Data were matched to the Target database using Proteome Discoverer 2.4 with parameters: trypsin digestion, static modification (Carbamidomethyl on C), dynamic modifications (M oxidation, N-terminal acetylation), precursor/fragment mass tolerances (±15 ppm/±0.02 Da), max 2 missed cleavages, and high confidence threshold (FDR < 0.01).

### 4.6. Inhibitory Effect of Recombinant Protein TasA on the Spores of Colletotrichum Acutatumfrom Rubber Trees

A conidial suspension of *C. acutatum* was prepared by harvesting spores from 5-day-old cultures grown at 28 °C. Spores were washed with sterile distilled water and filtered through three layers of sterile lens paper to remove hyphal debris, yielding a final concentration of 1 × 10^4^ CFU/mL. Aliquots of the suspension were treated with recombinant TasA fusion protein at two concentrations (30 μg/mL and 60 μg/mL), while untreated spores served as negative controls. Treated samples were incubated under static conditions at 28 °C for 24 h. Spore germination was assessed via light microscopy (400× magnification) at 2 h intervals, with germination rates quantified by counting germinated spores (≥50% germ tube elongation relative to spore diameter) across three biological replicates (minimum 100 spores/replicate).

### 4.7. Inhibitory Effect of Recombinant Protein TasA on the Spores of Oidium Heveae Steinmann from Rubber Trees

Wash off the spores of *O. heveae* Steinmann from rubber trees cultured at 28 °C for 5 days with sterile water and then filter them through three layers of lens paper to obtain the conidial suspension (with a concentration of 1–1.3 × 10^4^ spores/mL). Add 300 μL of the spore suspension into three 1.5 mL sterile centrifuge tubes, respectively. Treat the spore suspension with the TasA fusion protein at final concentrations of 30 μg/mL and 60 μg/mL. For the blank control, add sterile water. After static cultivation in an incubator at 28 °C for a certain period of time, conduct microscopic examination to observe the spore germination situation.

### 4.8. Determination of Inhibitory Effect of Recombinant Protein TasA on Plant Pathogenic Fungi

The antagonism experiment of recombinant protein TasA against pathogenic bacteria was carried out by plate confrontation method.

The PDA were poured into sterilized Petri dishes (inner diameter 60 mm), and the 50, 100, 150 μg/mL of recombinant protein TasA was added. No recombinant protein is used as the blank control. One mycelium plug (5 mm diameter) from the edge of four-day-old fungal pathogen (*C. acutatum* and *A. heveae*) was placed on the PDA in the center of the Petri dishes.The Petri dishes were sealed with parafilm and incubated in the dark at 28 °C for 4 days. Each treatment was tested in triplicate to confirm the reproducibility of results. First, the relative inhibition (%) of mycelial growth was determined using the following formula:(1)$$\mathrm{Mycelial}\;\mathrm{growth}\;\mathrm{inhibition}\;\left(\%\right)\;=\;\frac{C-T}{C}\;\times \;100$$
where C and Tare the average diameter (mm) of fungal mycelia in the control and treatment, respectively.

## 5. Conclusions

In this study, we cloned the *TasA* gene from the rubber tree-derived *B. amyloliquefaciens* BS-3 strain and compared it with proteins TasA from other *Bacillus* species. Notably, the BS-3-derived *TasA* gene was identified as the shortest reported to date with confirmed antifungal activity. Through prokaryotic expression using the junction-truncated vectors *pET28a* and *pCZN1*, we found that only the *pCZN1-TasA* construct yielded soluble recombinant protein in the supernatant of *E. coli* Arctic Express cells. The purified protein TasA exhibited significant antifungal activity against *C. acutatum* and *A. heveae*.

Our findings underscore the gene’s potential as a candidate for disease resistance applications, providing a theoretical foundation for its use in plant disease control. Additionally, these results further imply that biofilm formation may play a critical role in BS-3′s antagonistic mechanism against pathogens.

## Figures and Tables

**Figure 1 ijms-26-07529-f001:**
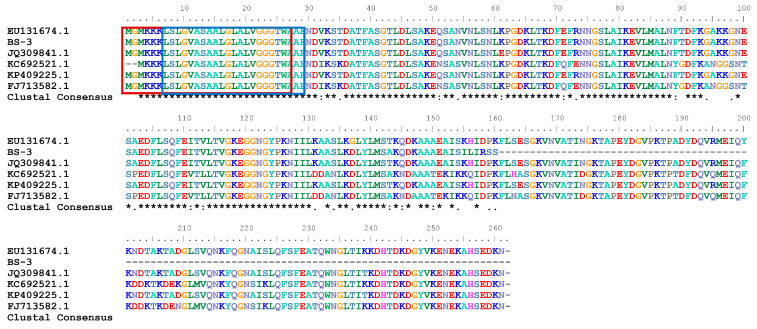
Multiple sequence alignment analysis of TasA. * (asterisk): represents a completely conserved position.

**Figure 2 ijms-26-07529-f002:**
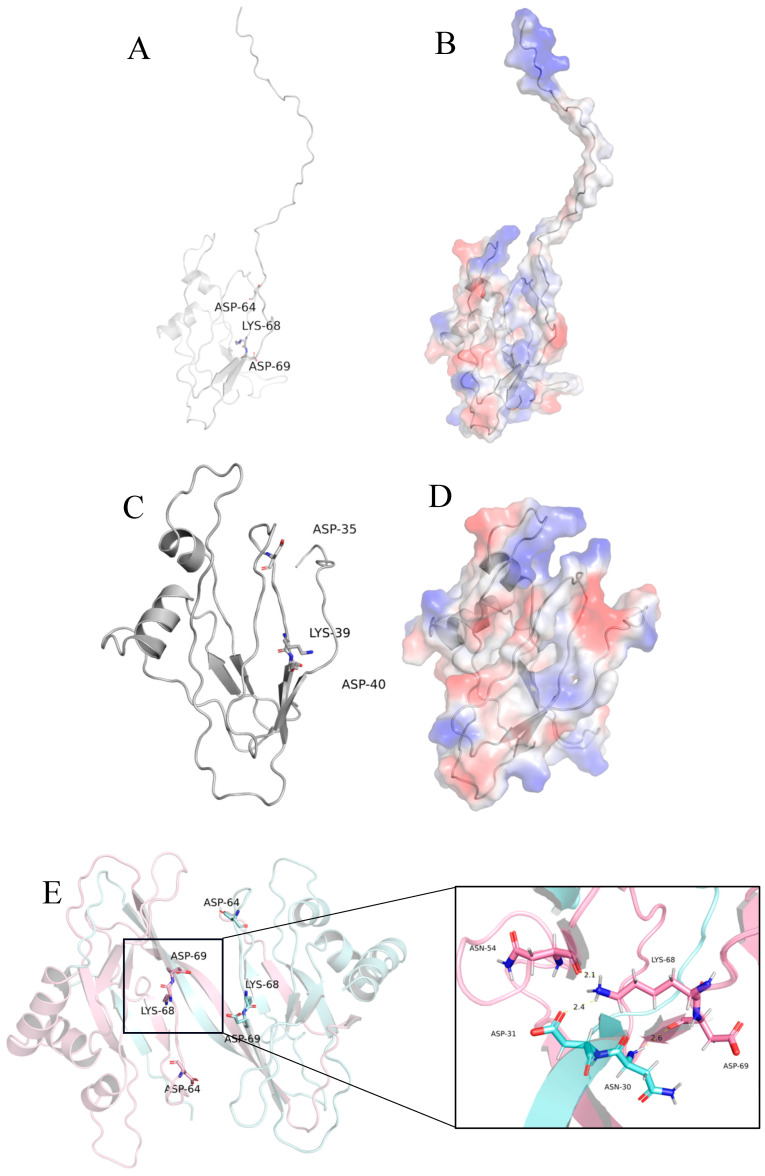
Three-dimensional structure prediction of *TasA* gene coding protein and the dimer conformation of the truncated TasA by AlphaFold3.Representative images: (**A**):Full-length cartoon, (**B**):full-length surface structure diagram, (**C**):truncation cartoon, (**D**):truncation-surface structure diagram, (**E**): use of AlphaFold3 to predict the dimer conformation of the truncated TasA.

**Figure 3 ijms-26-07529-f003:**
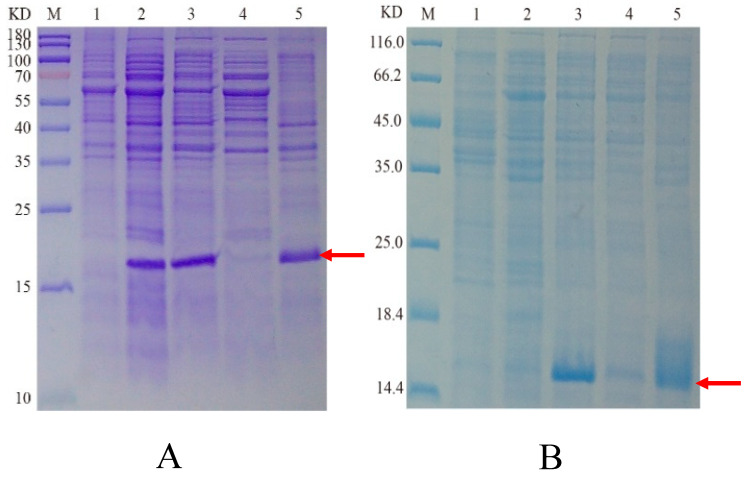
SDS-PAGE analysis of recombinant TasA. Representative images:(**A**): M:protein molecular weight marker; 1: total proteins of *E. coli* BL21(DE3) harboring the recombinant plasmid *pET28a*; 2: the uninduced *E. coli* BL21(DE3) total protein carries the recombinant plasmid *pET28a*-*TasA*; 3: induced cytoplasmic total protein; 4: supernatant after induced crushing; 5: precipitate after induced fragmentation. (**B**): M: protein molecular weight marker; 1: total proteins of harboring the recombinant plasmid *pCZN1*; 2: the uninduced total ArcticExpress protein carries the recombinant plasmid *pCZN1*-*TasA*; 3: induced cytoplasmic total protein; 4: supernatant after induced crushing; 5: precipitate after induced fragmentation.

**Figure 4 ijms-26-07529-f004:**
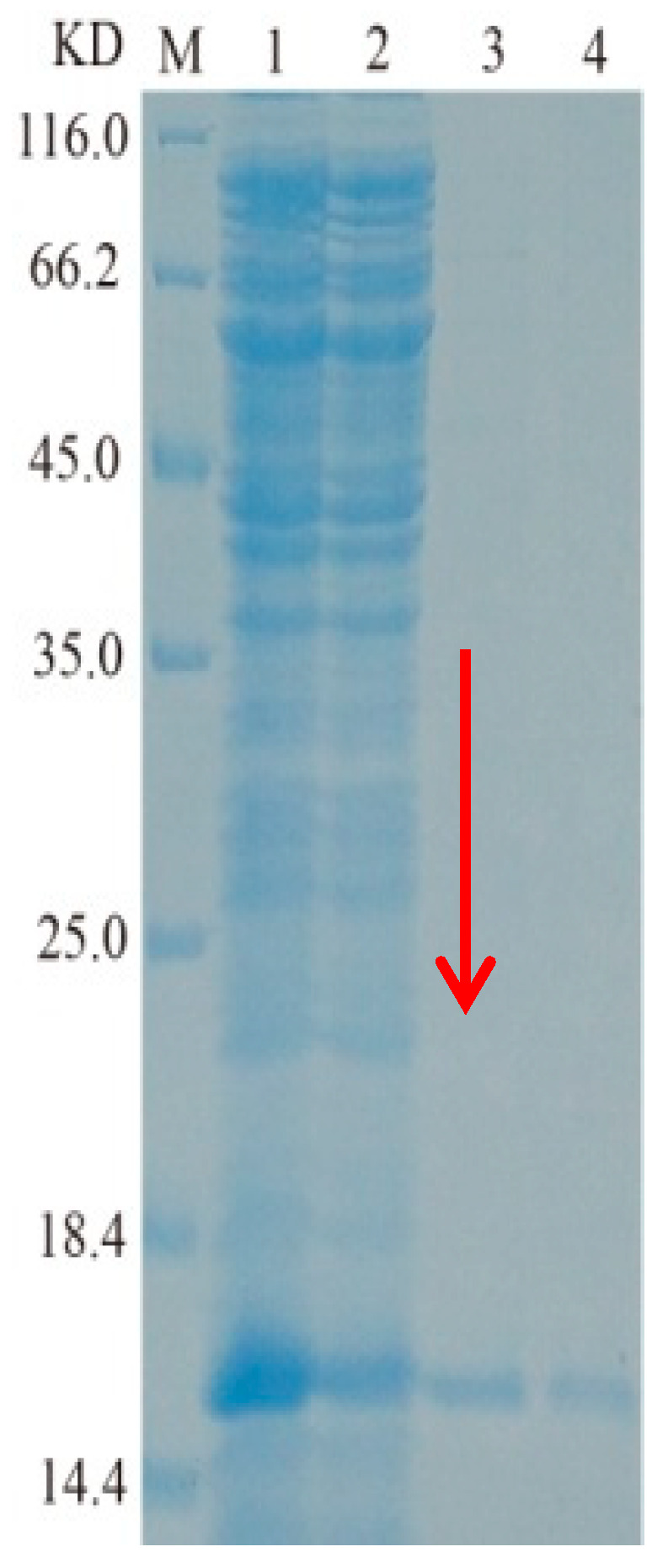
SDS-PAGE analysis of protein TasA. Representative images: M:Protein molecular weight marker; 1–2:induced cytoplasmic total protein; 3–4:purified TasA.

**Figure 5 ijms-26-07529-f005:**
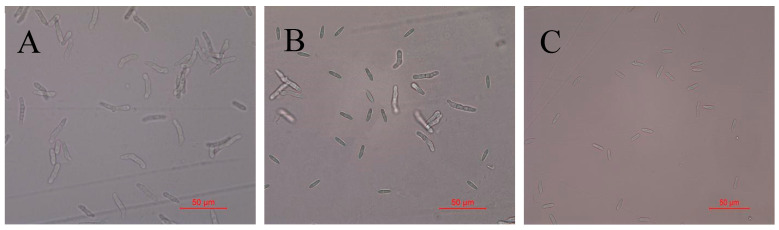
Conidial growth germination inhibition of *Colletotrichum acutatum* by the fused protein TasA for 12 h (magnification 400×). Representative images: (**A**): CK; (**B**): 30 μg/mL; (**C**):60 μg/mL.

**Figure 6 ijms-26-07529-f006:**
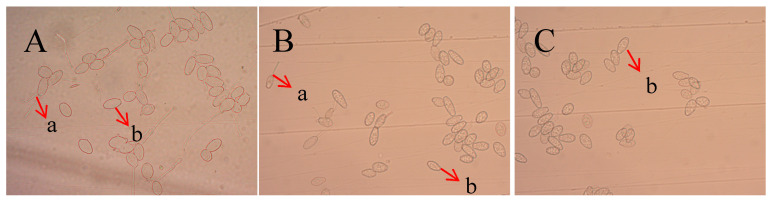
Spore germination inhibition of *Oidium heveae* Steinmann by the fused protein TasA for 6 h. Representative images: (**A**): CK; (**B**): 30 μg/mL; (**C**):60 μg/mL. a: germinated (healthy); b: not germinated (unhealthy).

**Figure 7 ijms-26-07529-f007:**
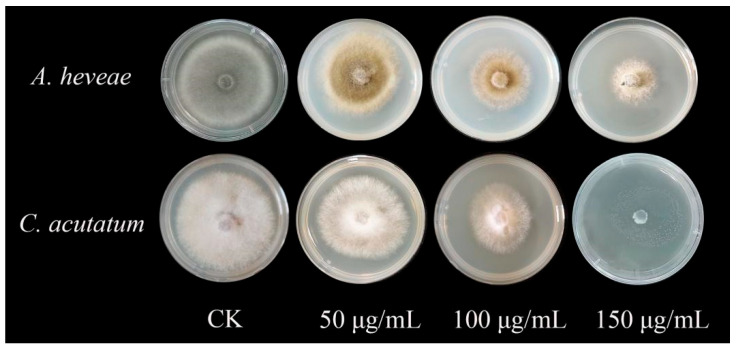
Antifungal activities of the fused protein TasA against mycelial growth of different pathogens.

**Table 1 ijms-26-07529-t001:** BLASTn comparison of *TasA* gene sequences was used to analyze the 7 gene sequences with the highest similarity.

	Number	GenBank	Query Coverage (%)	Similarity (%)
*B. amyloliquefaciens*	205	CP054415.1	100	99.18
	H	CP041693.1	100	99.18
	HK1	CP018902.1	100	99.18
	SRCM101267	CP021505.1	100	99.18
	SRCM123386	CP128501.1	100	99.18
	SRCM124317	CP116011.1	100	99.18
*B. subtilis*	ATCC 13952	CP009748.1	100	99.18

**Table 2 ijms-26-07529-t002:** The main structure of recombinant TasA and the fragments identified by mass spectrometry.

Sequence	Positions in Master Proteins	Theo. MH + Da	Charge: Sequest HT	*m/z* Da: Sequest HT
VHHHHHHMNDVK	5–16	1527.71346	3	509.91019
EQSANVNLSNLKPGDKLTK	33–51	2056.10331	3	686.04144
AASLKDLYLMSAK	115–127	1426.76103	3	476.25912
EVLMALNFTDFKGAK	65–79	1683.87745	2	842.44513
VHHHHHHMNDVKSTDATFASGTLDLSAK	5–32	3093.47077	4	774.12451
EGGNGYPKNIILK	102–114	1402.76889	2	701.88715
NNGSLAIKEVLMALNFTDFK	57–76	2225.16347	3	742.38464
QDKAAAEAISILIR	128–141	1498.85877	2	749.93365
NIILKAASLK	110–119	1070.69321	2	535.85052
GNESAEDFLSQFEITVLTVGK	81–101	2284.13433	2	1142.57556
QDKAAAEAISILIRSS	128–143	1672.92282	3	558.31409
EQSANVNLSNLKPGDKLTKDFEFR	33–56	2750.41078	4	688.35828
STDATFASGTLDLSAKEQSANVNLSNLKPGDKLTK	17–51	3621.86062	4	906.22137
STDATFASGTLDLSAKEQSANVNLSNLKPGDK	17–48	3279.63392	4	820.66321
MNHKVHHHHHHMNDVKSTDATFASGTLDLSAK	1–32	3645.71863	5	729.94092
DFEFRNNGSLAIK	52–64	1510.76487	2	755.88629
NIILKAASLKDLYLMSAK	110–127	1992.1562	3	664.72455
DLYLMSAKQDK	120–130	1327.65623	2	664.33289
EQSANVNLSNLKPGDK	33–48	1713.8766	2	857.44183
VHHHHHHMNDVK	5–16	1543.70837	4	386.68253
MNHKVHHHHHHMNDVK	1–16	2095.95622	3	699.32373
DLYLMSAK	120–127	940.48083	1	940.48022
EGGNGYPK	102–109	821.37881	2	411.19736
EVLMALNFTDFK	65–76	1443.71883	2	722.36908
STDATFASGTLDLSAK	17–32	1584.77516	2	792.8924
AASLKDLYLMSAKQDK	115–130	1781.9466	4	446.24292
NNGSLAIK	57–64	816.45739	2	408.73315
LTKDFEFR	49–56	1055.55202	3	352.52206
AAAEAISILIR	131–141	1127.67828	2	564.34387
DLYLMSAKQDK	120–130	1311.66131	2	656.33484
KGNESAEDFLSQFEITVLTVGK	80–101	2412.2293	3	804.75092

**Table 3 ijms-26-07529-t003:** Inhibition effect of TasA fusion protein on spore germination of *Colletotrichum acutatum*.

Concentrate/Time	4.5 h	6 h	12 h
CK	59.5% ± 0.36%	35% ± 0.15%	0% ± 0.00%
30 μg/mL	93.91% ± 1.35% **	91.95% ± 0.01% **	87.4% ± 0.42% **
60 μg/mL	100% ± 0.00% **	100% ± 0.00% **	100% ± 0.00% **

Note: Data are presented as the mean of three independent experiments. ** indicates an extremely significant difference compared with the control group at the same time point (*p* <0.01).

**Table 4 ijms-26-07529-t004:** Spore germination inhibition of *Oidium heveae* Steinmann by the fused protein TasA for 6 h.

Concentrate/Time	6 h
CK	38.75% ± 0.35%
30 μg/mL	88.38% ± 0.14% **
60 μg/mL	100% ± 0.00% **

Note: Data are presented as the mean of three independent experiments. ** indicates an extremely significant difference compared with the control group at the same time point (*p* <0.01).

**Table 5 ijms-26-07529-t005:** Antifungal activities of the fused protein TasA against mycelial growth of different pathogens.

Pathogen	Concentrate (μg/mL)	Diameter (mm)	Inhibition Rate (%)
*A. heveae*	150	31.29 ± 0.63	64.77 ± 1.34 **
	100	19.14 ± 0.14	39.62 ± 0.87 *
	50	12.62 ± 0.48	26.11 ± 1.26 *
*C. acutatum*	150	48.63 ± 0.15	98.6 ± 1.09 **
	100	23.07 ± 0.51	52.47 ± 1.19 *
	50	18.17 ± 0.24	36.83 ± 1.87 *

Note: Data are presented as the mean of three independent experiments. * indicates an extremely significant difference compared with the control group at the same time point (*p* <0.05). ** indicates an extremely significant difference compared with the control group at the same time point (*p* <0.01).

**Table 6 ijms-26-07529-t006:** Primers used in this study.

Primer	Restriction Enzyme	Sequences ^a^
TasA^S^		F: 5′-ATGGGTATGAAAAAGAA-3′
		R: 5′-AGAACTTCGGATCAATATGCT′-3′
*pET28a-TasA*	*EcoRΙ*	F:5′-CAAATGGGTCGCGGATCCGAATTCGCATTTAATGATGTGAAGTCCAC-3′
	*NotI*	R:5′-GTGGTGGTGCTCGAGTGCGGCCGCAGAACTTCGGATCAATATGCTGAT-3′
*pCZN1-TasA*	*NdeI*	F:5′-ATGAATCACAAAGTGCATCATCATCATCATCATATGAACGATGTGAAAAGCA-3′
	*XbaI*	R:5′-CTTTTAAGCAGAGATTACCTATCTAGATTAGCTGCTACGAATCA-3′

Restriction sites underlined. ^a^ The restriction endonuclease site in each oligonucleotide is underlined, and the enzyme that cuts it is listed at the right.

## Data Availability

No data was used for the research described in the article.

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
