# Peer review of "Cloning, Heterologous Expression, and Antifungal Activity Evaluation of a Novel Truncated TasA Protein from *Bacillus amyloliquefaciens* BS-3"

_ijms, 2025, doi:10.3390/ijms26157529_

Round 1
Reviewer 1 Report
Comments and Suggestions for Authors
The manuscript reviewed pertains to a current scientific field related to the analysis of genomic data of economically significant microorganisms, prediction of functionally important proteins, their production, structural analysis and detailed description of the spectrum of biological properties. Bacteria of the Bacillus genus are well known as objects of biological control in agriculture, as well as the basis for probiotic additives. The polypeptides with antimicrobial properties synthesized and secreted by them are the subject of active research. In this regard, the results of this work seem extremely significant and can be highly appreciated by the scientific community. During a careful review of this work, a list of questions and comments was formed, given below:
- In the Introduction section, the authors say that the TasA protein is important for the formation of biofilms in Gram-positive bacteria. However, it is well known that biofilm formation is one of the key factors of virulence and antibiotic resistance of Gram-negative forms. In this regard, the authors are invited to conduct a brief comparison of protein factors responsible for biofilm formation in Gram-negative and Gram-positive bacteria.
- In the same section, it is mentioned that the B. amyloliquifaciens strain was isolated from the roots of Hevea brasiliensis, which are very rich in rubber. In this regard, we would like to understand whether the genes of enzymes or any other functional polypeptides capable of binding and/or hydrolyzing this biopolymer have been annotated in this species?
- It remains unclear from the text how exactly the purified truncated form of the recombinant TasA protein was identified - by LC-ESI-MS/MS or MALDI-TOF MS? Please provide a detailed analysis methodology.
- For what reason were only two active concentrations (30 and 60 μg/ml) chosen for testing antimicrobial activity, rather than using, for example, two-fold serial dilutions. The fact is that with a protein molecular weight of about 17 kDa, the load is about 2 and 4 μM, respectively, which may be insufficient for a correct interpretation of the inhibition effect, which is usually estimated at least at 10 μM.
- The authors conclude that the TasA protein has homology with the Peptidase M73 superfamily group, that is, in fact, it can be an enzyme. Was the protease activity of this protein tested?
- Figure 1 - it would be good to highlight the signal peptide, transmembrane domain and mature sequence in the alignment as frames.
- Figure 2 - the authors are strongly recommended to carry out homology modeling of TasA with proteins from the Peptidase M73 superfamily, as well as with the already annotated in the PDB structures of TasA proteins from B. subtilis (PBD IDs: 5OF1 and 5OF2).
- Figure 3A – are there very few proteins in the inclusion body lysate (lane 4)? Please clarify, is this protein not detected at all in the water-soluble fraction?
- Figure 5 – it would be appropriate to transfer the mass spectra to Supplemental Data. Instead, it would be much clearer to provide the primary structure of recombinant TasA with an indication of the fragments that were identified by mass spectrometric analysis.
Reviewer 2 Report
Comments and Suggestions for Authors
Please see the attached document.

Round 2
Reviewer 1 Report
Comments and Suggestions for Authors
Tha authors have sugnificantly improved the manuscript, and described the most of questions applied. From my side, I'm ready to accept all answers, including the forth one. But I've got one more question based on the results presented in Tables 5-6. Is TasA able to be considered as potent antimycotic protein? If it is, I would recommend to provide additional testing according to the official CLSI of EUROCAST protocols to determine MIC values. You should use any metodology assay, for instance, in comparison with some commercial antifungal antibiotic compound.
Author Response
请参阅附件。

Reviewer 2 Report
Comments and Suggestions for Authors
Please see the attachment.
